# Polarization Switching in 2D Nanoscale Ferroelectrics: Computer Simulation and Experimental Data Analysis

**DOI:** 10.3390/nano10091841

**Published:** 2020-09-15

**Authors:** Ekaterina Paramonova, Vladimir Bystrov, Xiangjian Meng, Hong Shen, Jianlu Wang, Vladimir Fridkin

**Affiliations:** 1Institute of Mathematical Problems of Biology, Keldysh Institute of Applied Mathematics, RAS, Moscow 142290, Russia; ekatp@yandex.ru; 2National Lab. Infrared Physics, Shanghai Institute of Technical Physics, CAS, Shanghai 200083, China; xjmeng@mail.sitp.ac.cn (X.M.); hongshen@mail.sitp.ac.cn (H.S.); jlwang@mail.sitp.ac.cn (J.W.); 3Federal Center of Photonics and Crystallography RAS, Shubnikov Institute of Crystallography RAS, Moscow 117333, Russia; fridkinv@gmail.com

**Keywords:** LGD theory, polarization, nanoscale ferroelectrics, kinetics, homogeneous switching, computer simulation, fitting

## Abstract

The polarization switching kinetics of nanosized ferroelectric crystals and the transition between homogeneous and domain switching in nanoscale ferroelectric films are considered. Homogeneous switching according to the Ginzburg-Landau-Devonshire (LGD) theory is possible only in two-dimensional (2D) ferroelectrics. The main condition for the applicability of the LGD theory in such systems is its homogeneity along the polarization switching direction. A review is given of the experimental results for two-dimensional (2D) films of a ferroelectric polymer, nanosized barium titanate nanofilms, and hafnium oxide-based films. For ultrathin 2D ferroelectric polymer films, the results are confirmed by first-principle calculations. Fitting of the transition region from homogeneous to domain switching by sigmoidal Boltzmann functions was carried out. Boltzmann function fitting data enabled us to correctly estimate the region sizes of the homogeneous switching in which the LGD theory is valid. These sizes contain several lattice constants or monolayers of a nanosized ferroelectrics.

## 1. Introduction

Studies of polarization switching in ultrathin (nanoscale) polymer ferroelectric films of polyvinylidene fluoride-trifluoroethylene (P(VDF-TrFE)), obtained experimentally by the Langmuir-Blodgett (LB) method [1,2,3,4,5,6,7], have shown that for nanosized (within the film thickness, when their sizes are less than or equal to the critical sizes required for the formation of a domain core), homogeneous non-domain switching of polarization is observed [1,6]. This occurs in accordance with the Landau theory of phase transitions [8], developed for ferroelectrics by Ginzburg and Devonshire (Ginzburg-Landau-Devonshire (LGD) theory) [8,9,10]. In such homogeneous media, the kinetics of the process and the time of polarization switching are well described by the Landau-Khalatnikov equation [11,12,13], in the approximation of continuous homogeneous media.

This was shown experimentally [13,14,15,16] and theoretically (in first-principle calculations [17], including with molecular dynamics (MD) approaches [18] and using quantum–mechanical semi-empirical methods [17,18,19,20]). The main condition for the applicability of the LGD theory in such systems is its homogeneity along the polarization switching direction. 

It is clear that it is necessary to apply the numerical estimates obtained by the LGD theory with respect to specific and real ferroelectric samples (of any composition and geometry) with extreme caution, and it is necessary to take into account possible limitations and the approximate nature of the results obtained. Nevertheless, under certain conditions and for some structures, the estimates give quite reasonable values consistent with the experimental values, as we will see from the analysis of the results obtained for a number of different systems carried out in this article.

In this paper we do not deal with domain switching and consider only homogeneous switching and its transition to the domain one. But domain switching works well for the sizes when domain nuclei can already form—all these domain switching processes are remarkably described in the Tagantsev et al. monograph [21].

Naturally, many issues of domain formation and switching in various real systems remain quite complex and require careful analysis in each specific case. Note, in this case, that the phenomenological LGD theory itself does not answer all the questions and does not consider the mechanisms of polarization switching themselves (especially at the microscopic level). This is a continual theory and it describes the thermodynamics of changes in polarization during a phase transition, which is determined by the potential barrier between the polar and nonpolar phases in the LGD theory. This theory also considers the kinetics of changes in polarization in a homogeneous continuous medium according to the nonlinear Ladnau-Khalatnikov equations—again in the approximation of a homogeneous infinite medium. All this must be taken into account in applications to various real structures.

It is known that the LGD theory does not describe the switching of conventional bulk ferroelectrics, since it predicts the magnitude of the coercive field, which is 2 to 3 times higher than the experimental one. The large coercive fields predicted by the LGD theory came to be called *intrinsic* (or *proper*), and their experimental values are *extrinsic* (or *improper*).

Domain discovery helped resolve this inconsistency. As it turned out, such bulk ferroelectrics inevitably split into domains [21]. This division of the polar crystal into domains reduces the free energy of the crystal. In this case, the minimum of the free energy or thermodynamic potential of the crystal below the Curie point in the polar ferroelectric phase is achieved if the crystal is divided into domains.

The appearance of nanoscale ferroelectrics, namely, polymer ferroelectric films, the thickness of which reached the minimum possible values of one mono-molecular layer (~0.5 nm) [1,2,3,4,5,6,7], created a completely new and different situation.

The question of domain formation does not arise, since such layers are much smaller than the size of a possible domain (~10 nm [21]), but at the same time all these layers (as one layer and also all layers as a whole, if their formation is created from several layers) created significant spontaneous polarization in their polar phase. In this case, even the values of the coercive fields turned out to be much larger than those in bulk ferroelectrics and they were close to the values of the LGD theory. It became possible to talk about the applicability of the LGD theory here.

Thus, here and further in this article, we will not talk about the formation of domain nuclei and the growth of domains (and their role in polarization switching), but we are only talking about non-domain homogeneous polarization switching, under the conditions of a continuous homogeneous medium, when its size (thickness) does not exceed the dimensions of the formation of domains and one can speak of one continuous medium in the direction of polarization.

We also note that for some time it was still believed that such uniform switching is only possible in polymer ferroelectrics, and cannot be observed in other perovskite ferroelectrics and similar crystals. Nevertheless, since the discovery of homogeneous switching in ferroelectric polymers, it has become evident that all nanoscale ferroelectric films and crystals whose thickness is less than or comparable to the size of a domain nucleus can have a homogeneous nature of polarization switching. This has recently been proved by the example of ferroelectric films of a classical ferroelectric–barium titanate [1,3,22]—in their nanoscale form. In this work, a comparative analysis of such homogeneous polarization switching in some different nanoscale ferroelectrics is carried out.

## 2. Methods and Models

Before we move on to modeling and discussion of results obtained, we should make a short introduction into the LGD theory and its estimations and predictions. As it was written above, the values of the intrinsic coercive field *E_C_* are determined by the potential barrier between the polar and nonpolar phases as described by the polarization expansion coefficients of thermodynamic potential (or free energy) known from the LGD theory [4,5]:(1)$$\Phi =\Phi_{0}+\frac{\alpha }{2}P^{2}+\frac{\beta }{4}P^{4}+\frac{\gamma }{6}P^{6}-EP$$
where *Φ* is free energy, *P* is spontaneous polarization, *E* is an external field, and the coercive field *E_C_* is expressed as:(2)$$E_{C}=\frac{P_{0}}{\chi_{0}}f\left(t\right)$$
where *P*_0_
*= P*(*T = T*_0_) = √(-*β*/*γ*), χ_0_ = χ(*T = T*_0_) = *γ*/2*β*^2^, *f(t)* is the function:(3)$$f\left(t\right)=\frac{3}{25}\sqrt{\frac{3}{5}}\left(1-\frac{25}{24}t\right)$$
and the reduced temperature is:(4)$$t\;=\;\frac{4\alpha \gamma }{\beta^{2}}=\frac{4\gamma }{\varepsilon_{0}{C\beta }^{2}}\left(T-T_{0}\right)$$

Here, χ_0_ is the ferroelectric contribution to the dielectric susceptibility, and *α*, *β*, *γ* are coefficients of the expansion of free energy in even degrees of polarization, known from the LGD theory. For estimates, it can be assumed that approximately *E_C_ ~ P*/χ_0_
*~ P*/*εε*_0_, where ε is the relative permittivity and *ε*_0_ is the dielectric constant of the vacuum.

The polarization switching kinetics of two-dimensional (2D) polymer ferroelectrics was described by the Landau–Khalatnikov Equation [11] and its solution for first-order phase transitions in two-dimenstional ferroelectrics was considered in [12,13]:(5)$$\xi \frac{dP}{dt}=\frac{-\partial \Phi }{\partial P}=-\alpha P-\beta P^{3}-\gamma P^{5}+E$$
where *ξ* is the damping coefficient. In the general case, the gradient term can be taken into account. An investigation of the solution of this equation showed that in the vicinity of the coercive field *E_C_*, the switching time sharply increases, and its reciprocal can be expressed as [12]:(6)$$\tau^{-1}\approx \frac{1}{\tau_{0}}{\left(\frac{E}{E_{C}}-1\right)}^{\frac{1}{2}}$$
where *τ*_0_ ≅ 6.3*βγξ*/*β*^2^, otherwise:(7)$$\tau^{-2}\approx \frac{1}{\tau_{0}^{2}}\left(\frac{E}{E_{C}}-1\right)$$

In this case, *E_C_* is the proper (or intrinsic) coercive field of the ferroelectrics. This relation (7) shows the linear behavior of *τ*^−2^ along *E* in the vicinity of *E_C_* (for *E > E_C_*). This relation is more suitable for comparison with experimentally measured data and is used in [14,15,18]. This relationship turned out to be convenient for analyzing the results of theoretical calculations when modeling the polarization switching processes in polymer ferroelectrics by molecular dynamics (MD) methods [18,19].

The discovery of two-dimensional ferroelectrics [1,2] led to a new stage of polarization switching development, using the study of switching of these ultrathin single-crystal films. A start was made by the development of a new method for growing single-crystal ferroelectric polymer films of polyvinylidene fluoride-trifluoroethylene (P(VDF-TrFE)) [1,2,3,4,5,6,7].

To grow the ferroelectric films of this polymer, the Langmuir–Blodgett (LB) method [1,2,3,4,5,6,7,14,15,16,22,23,24] was used, based on the transfer of polymer chains (or mono-layers-ML) from the surface of the water to a substrate that carries an electrode. Figure 1a–d shows polymer chains and cells in the polar (ferroelectric) and nonpolar (paraelectric) phases (these images were built using the HyperChem 8.0 tool (Hypercube Inc., Gainesville, FL, USA) [20]); Figure 1e–g shows a transport scheme of LB layers and 1 ML observed in a scanning tunneling microscope. The Langmuir ferroelectric films obtained by this method in 1998 were thinner than any films that were previously obtained. The thickness of one monolayer (1 ML) was 0.5 nm; that is significantly less than the size of the critical domain nucleus known from the literature [21]. The thickness of Langmuir polymer films (two-dimensional ferroelectrics) was controlled by ellipsometry and atomic force spectroscopy, and two-dimensional ferroelectrics 0.5–1.0 nm thick were first obtained.

Experimental study [14,15] and computational simulation [17,18,19] of the polarization switching, carried out on these ultra-thin polymer ferroelectrics have confirmed the relationships (6) and (7). The MD approach calculations [18,19] were carried out using the HyperChem software [20], using various modes and semi-empirical methods (PM3, etc.) for correct quantum-chemical calculations at each step of the MD simulation run process. As an example of the MD run calculations, simple 2 PVDF chains affected by an external electric filed *E* are presented (Figure 2).

The final time, for example, for the case of the two-chain model rotation (switching time *τ_S_*) was estimated from these MD energy trajectories (see Figure 2c) using criteria [18]: *δ* = *E_K_*/*E_K_*_max_ < 10^–3^, where *E_K_* is the kinetic energy at the final point and *E_K_*_max_ is the kinetic energy at the maximum *E_KIN_* point of the chain rotation (as shown in Figure 2c). As a result, this corresponds to reaching the rest point of the rotating chain and its new position from another opposite orientation of the total dipole ***Dt*** vector and polarization vector. With a similar precision we calculated the values of the coercive field, obtained from a hysteresis loop, corresponding to the switching of the PVDF chain polarization into the opposite direction (or the same rotation of the PVDF chain). These data were further used in the calculations of the thickness dependence of the coercive field (see below in Section 3.1).

## 3. Results and Discussion

### 3.1. Polymer Ferroelectrics

As a result, this MD approach allows us to determine switching (or rotation) time *τ_S_* (Figure 2c) for any PVDF system as the applied electric field ***E*** varies. Using this approach, the switching time *τ_S_* = *τ* (Figure 3b) was calculated for various polymer ferroelectric models [18,19] and the results are compared with experimental data [14,15] (Figure 3a).

The critical behavior of *τ*^−2^ for *E → E_C_* for the P(VDF-TrFE) LB films was shown in [14,15] (see Figure 3a). Figure 3a presents the data measured for a film of 10 ML (5 nm). Dotted lines show theoretical points corresponding to Equation (7), and triangles show experimental data. For a thicker film of 30 ML (15 nm), the circles indicate the experimental values of *τ*^−2^, and the dotted line indicates the exponential dependence. Figure 3b shows the calculated data using the MD run for *τ*^−2^(*E*) behavior in a similar logarithmic scale, which qualitatively coincides with the experimentally established character of the behavior [14,15] (Figure 3a). The inset in Figure 3b shows a linear dependence of Equation (7) *τ*^−2^(*E*) on *E* near *E_C,_*_,_ enabling interpolation to the intersection point with the horizontal axis, which determines the value of the coercive field *E_C_* ~ 2.3–2.4 GV/m for 2–4 chain models (ML monolayers).

Taking into account that the field *E* is external and that for thin polymer layers representing monomolecular layers, the dielectric constant is *ε* ~ 2.4 [17,25,26,27,28,29] (while *ε* = 5 and greater is only for thick films), we obtain the limiting maximum value *E**_C_*_MAX_ ~ *E_C_*/*ε* ≅ 1 GV/m, which is a proper coercive field and corresponds to many known experimental data as well as the LGD theory [1,2,3,4,5,6,7,16,29].

A similar study of the P(VDF-TrFE) LB films of different thicknesses was carried out in [16] (Figure 4). It was shown that for small thicknesses of 2–6 nm, the coercive field *E_C_* is proper and practically unchanged, and in the region of thicknesses greater than 8 nm, a transition region arises. It was show that for thicknesses greater than 10–12 nm, the proper coercive field *E_C_* becomes improper and is determined by the domain mechanism.

Another method for obtaining a coercive field is by calculation of hysteresis loops *P*(*E*) [27,28]. Calculations performed for different numbers of chains and film thicknesses using both these methods showed that the dependence of the obtained coercive field [18,19,28] is in good agreement with the experimentally established dependence of the coercive field *E_C_* on the film thickness [16] (Figure 5).

This dependence can be conditionally divided into 3 regions: the region of purely homogeneous LGD switching (up to 8 nm); the transition region (8–12 nm), where a kind of domain precursor is noted; and the region above 12–16 nm and further, where the domain switching mechanism predominates (the coercive field remains almost unchanged and is kept at a low level of ~0.07–0.05 GV/m) [16,18,19,27,28]. Thus, these switching calculations of ultrathin polymer ferroelectrics confirm that two-dimensional ferroelectrics can consist, in principle, of several monolayers or several unit cells. These data are also in good agreement with the results for thin BaTiO_3_ films (see in Figure 6 below), in which homogeneous switching was recently found at the scale level up 10 nm, and with a further increase in thickness they already correspond to thick films [1,2,3,4,22].

### 3.2. Barium Titanate

The switching kinetics in these ultrathin single-crystal BaTiO_3_ epitaxial laser films of 2–8 and 40 nm thickness synthesized on a SrRuO_3_/SrTiO_3_ substrate were studied in a SrRuO_3_–ITO capacitor using an atomic force microscope. A detailed description of the technique is presented in [30]. The distinction between proper and improper behavior is shown in Figure 5, which shows the dependence of the switching time *τ* on the applied voltage V for a film of 8 nm thickness in a capacitor (Figure 6a), and under a tip of an atomic force microscope (Figure 6b). For comparison, the same dependence for a bulk BaTiO_3_ single crystal of 1 mm thickness is shown in Figure 6c.

The dependence *τ*^−2^(V) for a barium titanate film in a capacitor and in an atomic force microscope is shown to be consistent with formula (2) (Figure 6a,b). These experimental results agree well with relation (7). On the contrary, the switching kinetics of the bulk crystal (Figure 6c) is nearly exponential; the switching has a domain nature. Figure 6d shows the dependence of the coercive field on the film thickness of barium titanate in the range from 2 to 40 nm. It is seen that in the range from 2 to 10 nm, the coercive field is intrinsic and proper (*E_C_* ~ 0.12 GV/m) and weakly depends on the thickness, in accordance with the LGD theory. For thicknesses of 40 nm (or more), the coercive field sharply decreases, which corresponds to the transition to domain switching. Correspondingly, films with a thickness of 2 and 8 nm exhibit their intrinsic homogeneous LGD switching (5)–(7), and thicker films exhibit a domain exponential dependence. These data are also in good agreement with the above results for LB polymer films [16] (Figure 4 and Figure 5).

Thus, the obtained and experimentally observed dependencies of the switching time on the thickness of the barium titanate film (as well as the polymer films mentioned and described above) fully correspond to the Landau-Khalatnikov kinetics, and give the values of the coercive field *E**_C_*, which exactly follows from the phenomenological LGD theory.

At the same time, it should be noted that the surface field caused by the surface charge, in principle, affects the coercive field in thin two-dimensional and nanoscale ferroelectric fields [31]. However, as it was shown in paper [32], the switching kinetics of these two-dimensional and nanoscale films exactly follow the Landau-Khalatnikov kinetics. In addition, this has been verified in experiments [14,15,22] and demonstrated in this work for ferroelectric polymer films and perovskite ferroelectric films.

Moreover, it should be reiterated, that it is Landau-Khalatnikov kinetics that give the exact eigenvalue for the values of the coercive field *E**_C_* for nanosized ferroelectrics, which follows from the phenomenological LGD theory.

### 3.3. Nanosized Materials Based on Hafnium Oxide

In the past few years, there has also been interest in nanoscale films based on hafnium oxide [33,34,35,36], including those doped with silicon and several types of their solid solutions, such as for example, Hf_0.5_Zr_0.5_O_2_ (HZO) [34,35]. Ferroelectricity was found in them and the corresponding hysteresis loops (Figure 7) suggest the proper intrinsic nature of the coercive field (~0.1–0.2 GV/m) here and, apparently, the homogeneous nature of the polarization switching. Though these samples are fully blended HZO solid solutions [36], there are some doubts as to whether they are fully homogeneous and fully monocrystallic, since they can consist of several phases. Therefore, it is too early to draw conclusions about homogeneous switching and its kinetics; these studies are ongoing.

The dependence of the coercive field on the thickness of the ultrathin HZO films (4–20 nm) is also observed (some data are given in review [36]). However, there are no reliable and detailed measurements yet. Nevertheless, the study of these nanoscale films is of great interest and here we also tried to estimate whether their possible parameters match with homogeneous polarization switching. For example, one can estimate the data of [35] (Figure 7); at least approximately, they correspond to the dependence of the coercive field EC on the films’ thickness x in such HZO-based films.

### 3.4. Boltzmann Function Fitting Data

Relying on the fact that the character of the transition region has a pronounced sigmoidal type for all the above-considered samples, we carried out fitting of various available experimental data in accordance with the formula of the sigmoidal Boltzmann function (in the form used by the OriginLab software, OriginLab Corp., Northampton, MA, USA) [37]:(8)$$E_{C}=\frac{A_{1}-A_{2}}{1+\exp\left(\frac{x-x_{0}}{dx}\right)}{+A}_{2}$$
where *A*_1_ and *A*_2_ are parameters of the maximum and minimum function values (corresponding to the values of the coercive field *E_C_*—a proper (intrinsic) and improper (extrinsic), respectively, *x_0_* is the average thickness of the films or samples corresponding to the middle of the transition region, and *dx* is the effective half-width of the transition region (see in Table 1).

The results obtained generally show that Equation (8) describes rather well how a coercive field changes as the thickness of all the samples varies. Moreover, for PVDF, the values obtained from the experiment and those obtained by MD modeling and hysteresis loops (especially the left side) calculations are close (see in Figure 5).

These values of the coercive field turn out to be the highest of all the experimental samples and indicate the existence of an intrinsic coercive field *Ec* up to sizes of at least 5–8 nm (which corresponds to 10–16 ML, where 1 ML = 0.5 nm [1,2,3,4,5,6,7]). That is, in this case, the existence of two-dimensional ferroelectrics is obvious up to ~10–16 PVDF ML (or up to ~8 nm).

At the same time, it turns out to be rather unexpected that for BaTiO_3_ the inflection point of the sigmoid (8) here occurs at *x_0_* ~ 19–20 nm, that is, here the region of existence of a homogeneous ferroelectric in this perovskite crystal structure is almost 3 times greater than that of a polymer PVDF film. For HZO, the order of the width of the region is ~5 nm, which is rather close to that of PVDF, but with a significantly larger half-width *dx ~* 6 nm, meaning there exists “smearing” of the transition region in a twice wider range. This, of course, is due to the insufficient number of accurate experimental data. It should be noted that the maximum value of *Ec* for barium titanate and HZO-based films turns out to be significantly lower than PVDF (almost by an order of magnitude). This, in principle, is not surprising and would be expected.

Thus, the proposed approach shows satisfactory fitting data and can be extended to the analysis of other similar data. In addition, it can be used to obtain significant parameters for two-dimensional ferroelectrics and homogeneous polarization switching in them.

## 4. Conclusions

The study of nanoscale ferroelectrics became possible for the first time when polymer ferroelectric films were synthesized by the Langmuir-Blodgett method [1,2,3,4,5,6,7]. This led to the discovery of two-dimensional ferroelectrics [2]. Soon, for classical perovskite ferroelectrics such as barium titanate, it became possible to create, by laser epitaxy, films several tens of nm thick [22]. The study of nanoscale perovskite films of barium titanate showed that they can also be two-dimensional ferroelectrics [1,2,3,4]. It can be assumed that other nanoscale single crystals can be two-dimensional and homogeneous. Recently, such nanoscale films have been obtained on the basis of hafnium oxide [33,34,35,36].

In this case, the phenomenological LGD theory describes well the switching kinetics, only if the medium is homogeneous and the size is slightly less or of the order of the critical size of the domain nucleus formation, which is the case for two-dimensional ferroelectrics. Thus, by two-dimensional ferroelectric we mean a nanoscale crystal, which in the direction of its switching can be considered to be homogeneous. It can consist, in principle, of several unit cells (or monolayers). Molecular modeling and quantum-mechanical calculations of polymer ferroelectrics using hysteresis loops and molecular dynamics methods were in good agreement and the presence of a transition region from homogeneous switching to domain switching of several nm (6–20 nm) length, thereby convincingly confirming that two-dimensional ferroelectrics can have several monolayers and cells. Similar results were obtained for nanoscale barium titanate and hafnium oxide.

The sigmoidal nature (described by the Boltzmann function (8)) of the transition region turned out to be common here. For various nanoscale ferroelectrics, it is possible to approximate quite effectively and thereby determine the important parameters of the transition region from the maximal value A1 (corresponding to the proper intrinsic coercive field *E_C_* for nanoscale films) to the minimal A2 value (corresponding to the known improper coercive field in thick films and bulk crystals) with half-width *dx* and middle point *x_0_*. These parameters, in general, determine the region of homogeneous switching in such nanosized ferroelectrics.

Therefore, one should distinguish between the intrinsic coercive field *E_C_* for nanoscale homogeneous ferroelectric films and the previously known improper coercive field *E_C_* in thick films and crystals, associated with the domain mechanism. The intrinsic field is several orders of magnitude greater than the experimental (improper) one.

In fact, in nanoscale ferroelectric films there is a competition between these two switching mechanisms: homogeneous and domain. As a result, when the film thickness increases, the domain mechanism should prevail. Thus, this parameter *x*_0_ actually shows and estimates the size of the possible region of validity of the existence and applicability of the LGD theory for homogeneous polarization switching in the nanoscale ferroelectrics.

## Figures and Tables

**Figure 1 nanomaterials-10-01841-f001:**
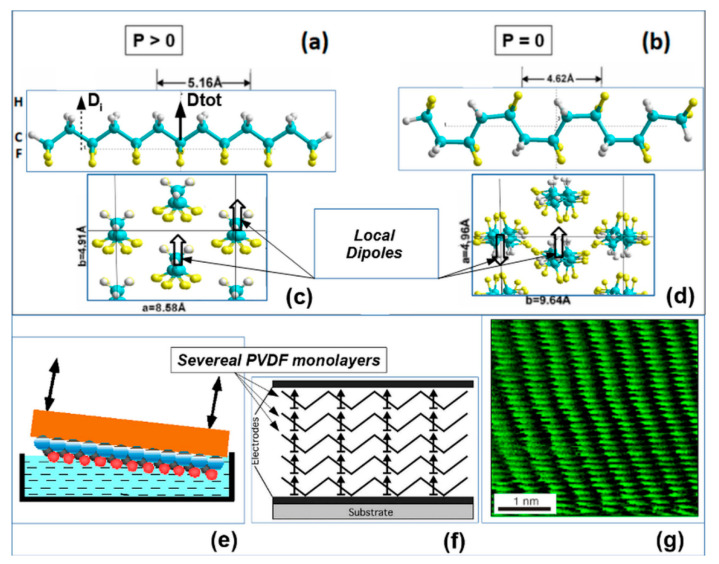
Ferroelectric polymer polyvinylidene fluoride (PVDF): (**a**,**c**) PVDF in polar trans conformation and total polarization *P* > 0; (**b**,**d**) PVDF in a nonpolar gauche conformation with total polarization *P* = 0. Reproduced with permission from [23]; AIP Publishing, 2012. (**e**) The formation of the PVDF Langmuir-Blodgett (LB) film on the surface of the water. Reproduced with permission from [4]; IOPscience, 2000. (**f**) Transferring several mono-layers (ML) of LB PVDF film onto a substrate with an electrode. Reproduced with permission from [24]; IEEE, 2005. (**g**) An image of the 1 M LB film of polyvinylidene fluoride-trifluoroethylene P(VDF-TrFE) by scanning tunneling microscopy [1,2]. Reproduced with permission from [2]; Springer Nature, 1998.

**Figure 2 nanomaterials-10-01841-f002:**
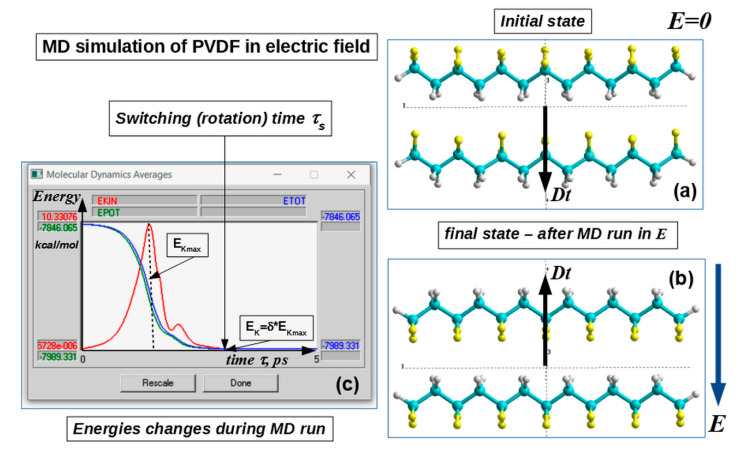
Scheme of the MD run process for a 2 PVDF-6 chains model with PM3 (in restricted Hartree-Fock approximation–RHF) calculations at each MD run step: (**a**) initial state; (**b**) final state after the MD run with dipole moment D orientation turned (switched) in the opposite direction in the electric field E; (**c**) changes of the MD average energies trajectory over time during the MD run (in ps) and the time of the switching *τs* for this PVDF system (red–kinetic energy, blue–total energy, green–potential energy). Reproduced with permission from [18]; Elsevier, 2014.

**Figure 3 nanomaterials-10-01841-f003:**
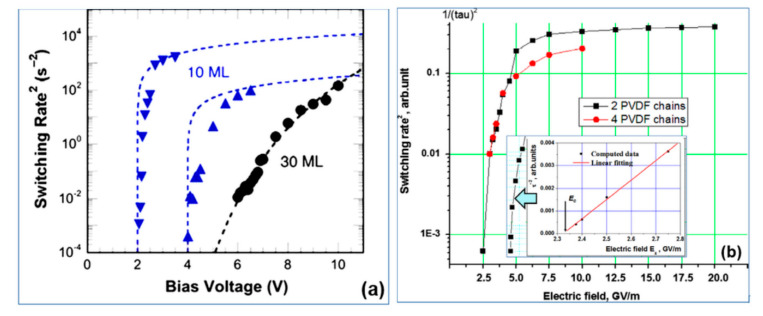
Switching time *τs* for polymer ferroelectrics presented as *τ*^−2^ along *E* in a logarithmic scale: (**a**) experimental data [14,15] for thin LB of P(VDF-TrFE) film of 10 ML (5 nm) with critical behavior of *τ*^−2^ for *E → E_C_* in comparison with thick 30 ML (15 nm) films with exponential low (Reproduced with permission from [14]; AIP Publishing, 2011); (**b**) MD simulation run data [18,19] for PVDF models showing the critical behavior of *τ*^−2^ with a change in *E*, which qualitatively coincides with the experimentally observed behavior; the inset shows the linear behavior of *τ*^−2^ as *E → E_C_* and determines the critical value of *E_C_* ~ 2.3 GV/m (Reproduced with permission from [18]; Elsevier, 2014).

**Figure 4 nanomaterials-10-01841-f004:**
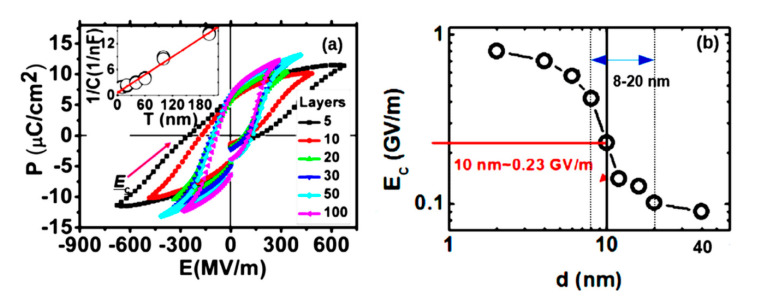
Hysteresis loops of the LB PVDF film [16]: (**a**) with different numbers of monolayers: 5, 10, 20, 30, 50 and 100 (the inset shows the linearity of the reciprocal capacity depending on the thickness); (**b**) *E_C_* as a function of the LB thickness of the PVDF film (with the transition region at 8–20 nm). Reproduced with permission from [16]; AIP Publishing, 2014.

**Figure 5 nanomaterials-10-01841-f005:**
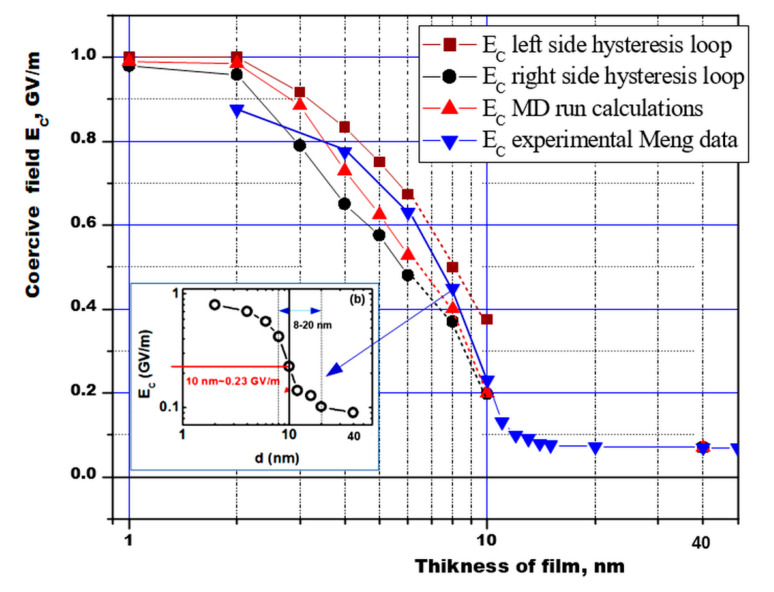
The dependence of the coercive field of the ferroelectric polymer PVDF on the film thickness, according to the results of calculations by different methods (from hysteresis loops and MD runs) in comparison with experimental data [16] (inset (b) is from Figure 4), taking into account the dielectric constant of an ultrathin molecular film *ε* ≈ 2.4. (It is our new original recalculated data, which continues from our preliminary calculations and estimations [18,19,27,28]).

**Figure 6 nanomaterials-10-01841-f006:**
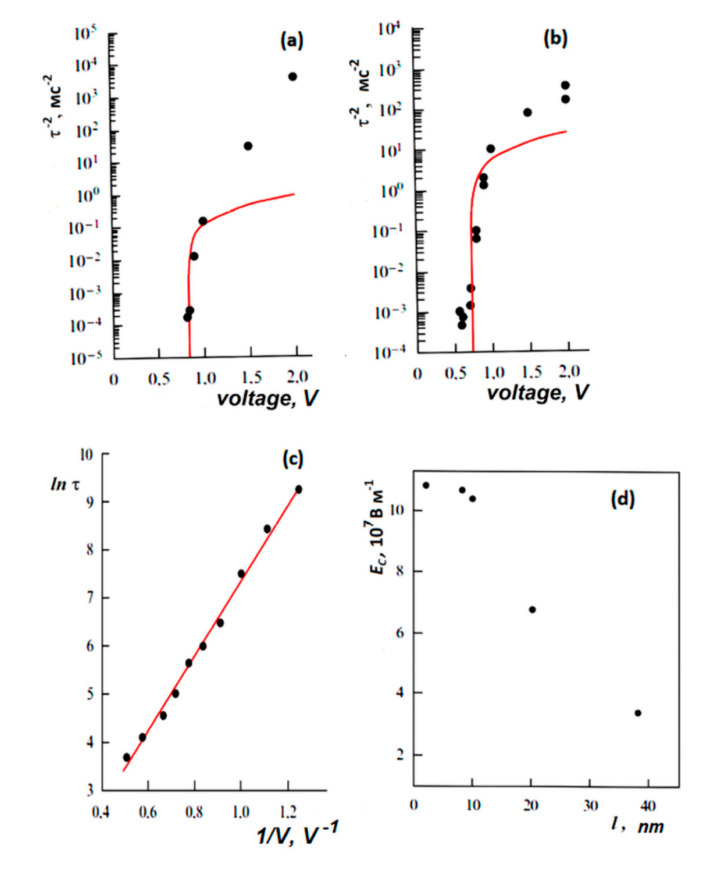
The dependence of the switching time *τ* on the applied voltage for 8-nm-thick BaTiO_3_ film [1,3,22]: (**a**) in the capacitor and (**b**) in the probe mode in an atomic force microscope (lines *τ*^−2^ (V) on (**a**) and (**b**) correspond to the calculations according to formula (7)); (**c**) for a BaTiO_3_ crystal, the line corresponds to ln[*τ*(V^−1^)] and is obtained relying on the piezo-response mode in an atomic force microscope; (**d**) the dependence of the coercive field on the thickness of the BaTiO_3_ film. Reproduced with permission from [22]; Elsevier, 2013.

**Figure 7 nanomaterials-10-01841-f007:**
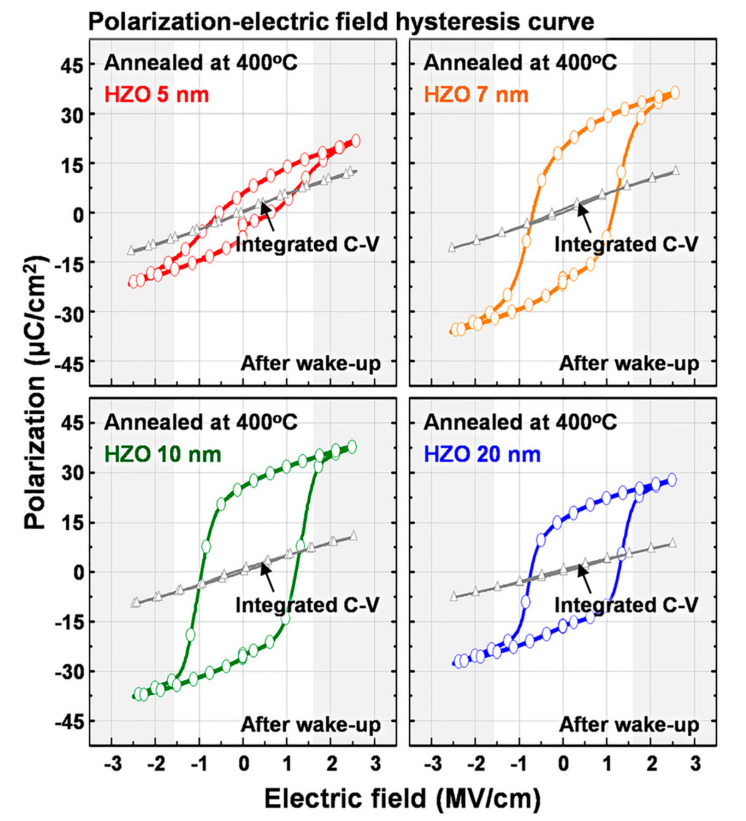
Polarization-electric field hysteresis and integrated capacitance voltage hysteresis curves of 5-, 7-, 10-, and 20-nm-thick HZO-based metal–insulator–metal (MIM) capacitors after wake-up field cycling (Reproduced with permission from [35]; AIP Publishing, 2018).

**Table 1 nanomaterials-10-01841-t001:** Fitted parameters of the Boltzmann function for various nanoscale ferroelectrics in comparison with modeling and calculations using MD runs and hysteresis loop data.

		Parameters of Boltzmann Function Fitting				
		A_1_, GV/m	A_2_, GV/m	*dx*, nm	*x_0_*, nm	Accuracy
Modeling	MD [18,19]	1.28 ± 0.10 1.26 ± 0.10 1.22 ± 0.14	0.067 ± 0.023 0.067 ± 0.024 0.075 ± 0.038	3.12 ± 0.37 3.09 ± 0.38 2.67 ± 0.51	5.10 ± 0.69 5.21 ± 0.61 5.02 ± 0.77	0.00052 0.00056 0.00152
	MD run (av.)	1.25 ± 0.20	0.070 ± 0.051	2.96 ± 0.74	5.11 ± 1.19	
	Left HL *^)^	1.23 ± 0.07	0.072 ± 0.019	3.42 ± 0.36	6.30 ± 0.51	0.00035
	Right HL *^)^	1.51 ± 0.38	0.068 ± 0.038	3.29 ± 0.75	2.99 ± 1.82	0.00148
Experiments	PVDF [16]	0.89 ± 0.03	0.062 ± 0.009	1.71 ± 0.14	7.55 ± 0.19	0.00039
	BaTiO_3_ [3,22]	0.11 ± 0.00005	0.035 ± 0.00004	3.83 ± 0.017	19.13 ± 0.012	1.668 × 10^−9^
	HZO [30]	0.13 ± 0.04	0.055 ± 0.001	5.77 ± 1.66	4.61 ± 5.52	1.399 × 10^−6^

*^)^ Data obtained from calculations of the hysteresis loop (HL) [27,28]: Left HL is based on the *E_C_* corresponding to the left side of HL, Right HL is based on the *E_C_* that was obtained from the right side of HL.

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
