# Peer review of "Polarization Switching in 2D Nanoscale Ferroelectrics: Computer Simulation and Experimental Data Analysis"

_nanomaterials, 2020, doi:10.3390/nano10091841_

Round 1

Reviewer 1 Report

Dear Authors:

I have looked through this manuscript again. In my opinion, all comments from me have been resoled in the revised version. Therefore, I suggest that this paper should be accepted in the current form.

Author Response

Response to Reviewer 1

We agree and grateful to reviewer 1 for his opinion that this our revised paper should be accepted for publication in the current form.

We also grateful tor reviewer 1 for his earlier comments of the initial version of this paper, which allow us to improve it. 

Reviewer 2 Report

The figures are now better labeled but still formated in completely different styles. The text was revised but still doesn't flow that well.

I still wonder how this paper is benefiting the scientific community.

I would revise the scope of the paper but I guess that goes beyond the review process. Given that scope, I guess it just passes.

Author Response

Response to Reviewer 2

Our full responses to Reviewer 2 - in more detailed attached file

Reviewer 3 Report

The revised manuscript can be publishable.

Author Response

Response to Reviewer 3

We grateful to Reviewer 3 for his opinion: 

"This revised manuscript can be publishable"

We also grateful to Reviewer 3 for his previous comments, which allow us to improve this manuscript for publication.

This manuscript is a resubmission of an earlier submission. The following is a list of the peer review reports and author responses from that submission.

Round 1

Reviewer 1 Report

This paper investigated the polarization switching kinetics of nano-scale ferroelectric crystals, such as P(VDF-TrFE), BaTiO3, and (Hf,Zr)O2, especially the transition from homogeneous to domain switching. After carefully reading the manuscript, I do not think that this manuscript can be published in this journal. My comments are listed as followed:

  1. There are many errors or correct types in the figures. In FIG 1, the cited references should be in figure caption, not the figure. In FIG 2, the inset picture has not any index in X and Y axis, liking the unit. Similar issue in FIG 6.
  2. In FIG 3, I am very confused about the experimental data, which were obtained from the reference?
  3. The author claims that "Taking into account that the field E is external and that for thin polymer layers representing monomolecular layers, the dielectric constant ε is ~ 2.4 [17, 23, 24] (while ε is for thick films)". Generally, the dielectric constants of P(VDF-TrFE) should be in the range from tens to one hundred.
  4. The formation of reference section is not uniform. For instance, "16. Wang, J.L.;"; "22...switching in ultrathin ferroelectric BaTiO3 films. - arXiv:1204.4792 [cond-mat.mes-hall], 2012."
  5. Again, there is not any page number in the manuscript.
  6. Badly, many grammar problems in this manuscript. In Page 1 Line 34, "In such homogeneous media, the kinetics of the process and the switching time of polarization switching is well described"; "This was shown both experimentally [13-16] and theoretically"; "Before consider modeling and results "; "In this case, EC is the proper (intrinsic) coercive field ECint of the ferroelectric."; "Experimental study [14, 15] as well computational simulation [17-19] of the polarization switching";"A detailed technique  can  be  found  in  [27].", etc.
  7. The introduction and the conclusion parts seems lengthy and lacking any intrinsic interest.

Reviewer 2 Report

The authors report on the polarisation switching in nanoscale ferroelectric films, made from a polymer and BTO and HZO.

The main conclusion is that the "phenomenological LGD theory describes well the switching kinetics if only the medium is homogeneous and the size is slightly less or of the order of the critic[a]l size of the domain nucleus formation" and "In fact, in nanoscale ferroelectric films there is a competition between these two mechanisms. As a result, when the film thickness increases, the domain mechanism should prevail."

First impression:
- the figures stand out. Every figure looks completely different, some are in Russian and without units, while others are of low resolution or screenshots. I would strongly suggest making them more consistent and readable.
- the equations may be better rendered using in a style more consistent with the journal.
- there are quite a few sentences that are not fully correct.

These formal aspects can be fixed and the journal may have a more succinct opinion on that.

Structure:
The introduction is fairly standard but does not quite motivate why these three materials are used in particular or what questions shall be answered. As a consequence, it is hard to follow the paper as the three sections are not well linked and complementary. Some editing could help here to help the reader to better grasp the connections. An example of one such connection: the calculations in figure 2 should be discussed with the data in figure 3.

Reproducibility:
The paper does not sufficiently describe the methods and the experiments to be repeated by an independent group.

Novelty:
It seems that most of the work is based on published research from 2014 and that many of the findings have been published before. It is indicated by references but it could be a bit clearer what part is previous work and what is new through this paper. I have thus difficulties to discern the advancement. Especially, the conclusions (quoted above) are basically a rephrasing of the facts in the introduction. It is nice to confirm and challenge the status quo but I feel the paper clarifies this if this is the purpose of the paper.

The authors have a long publishing record and have contributed to the field. I feel that this publication does not quite live up to their status in the field. I cannot recommend this paper for publication, based on lack of novelty and the lackluster form of the current manuscript.

Due to my conclusion, I refrain from commenting in greater detail on certain sentences or interpretations.

Reviewer 3 Report

The authors studied the polarizations switch in 2D nanoscale ferroelectrics. The main claims are the following: (i) E_c’s magnitude to indicate if the switch is homogeneous (nano-sized) or domain-formation (bulky), and (ii) fitting by sigmoidal Boltzmann functions.

The English presentation is poor (for instance, Lines 95~96) and there are inconsistent symbols in the equations (compare symbols in Line 108~110 with those in equation (2)). What is more critical is that there is a major flaw in logic.

The authors claim that the small thickness of the ferroelectric film guarantees the large E_c which is the indicator of homogenous switch. However, it is well known [Kretschmer, R. & Binder, K., surface effects on phase transtions in ferroelectrics and dipolar magnets. Phys. Rev. B 20, 1065 (1979)] that for a small thickness the surface effect, i.e. the depolarization due to surface charge P\dot Normal_vector, is critical. Such a critical surface effect is completely ignored in all considerations throughout the paper. Thus, publication of the manuscript is not recommended.